# L(M)V-IQL: MULTIPLE INTENTION INVERSE REINFORCEMENT LEARNING FOR ANIMAL BEHAVIOR CHARACTERIZATION

## ABSTRACT

In the pursuit of comprehending decision-making, behavioral neuroscience has made significant progress, aided by mathematical models in recent years. Among various approaches, Inverse Reinforcement Learning (IRL) stands out as a promising technique, distinguishing itself from other paradigms through its ability to circumvent the necessity for a reward function in characterizing observed behavior. Nevertheless, the widespread adoption of IRL within the field of neuroscience remains limited. This constraint may be attributed, in part, to the prevailing assumption in many existing IRL frameworks that animals exhibit a singular intention throughout a given task, wherein their behavior is optimized based on a single static reward function. In an effort to overcome this limitation, we propose the class of *Latent (Markov) Variable Inverse Q-learning (L(M)V-IQL)* algorithms, a novel IRL framework designed to accommodate multiple discrete intrinsic rewards. We formulate an Expectation-Maximization approach to cluster observed trajectories into multiple intentions, and subsequently solve the IRL problem independently for each intention. We illustrate the application of L(M)V-IQL through simulated experiments, followed by its utilization on a dataset of mice engaged in a two-armed bandit task. Our methods exhibit exceptional proficiency in discerning animal intentions and yield interpretable reward functions corresponding to each identified intention. We anticipate that this progress will open up new possibilities in neuroscience and psychology, serving as an important advancement in elucidating the intricacies of animal decision-making and uncovering underlying brain mechanisms.

## 1 INTRODUCTION

The characterization of animal's decision-making behavior stands as a fundamental objective within the field of behavioral neuroscience (Niv, 2009; Wilson & Collins, 2019). Prior research has formulated a variety of mathematical behavioral models across diverse tasks, encompassing generalized linear models and models based on reinforcement learning (Ashwood et al., 2022b; Beron et al., 2022). Such *forward models* facilitate the comprehension and comparison of decision-making strategies employed by both human and animal subjects. Additionally, they offer a low-dimensional behavioral representation suitable for regression analysis with neural activities (Hattori et al., 2019; Hamaguchi et al., 2022). Forward models require a empirically defined reward function that guides subjects optimizing their behavior during decision-making. However, defining a comprehensive and suitable reward function can pose challenges in complex behavioral tasks. Alyahyay et al. (2023) introduced a response-preparation task where subjects ought to hold a lever until a cue indicating the release signal. In this task, subjects can receive a binary extrinsic reward from the environment, whereas the intrinsic reward driving behavior such as hunger, thirst, engagement, associated with each timestamp is, however, not obvious to the experimenter. As another example, within a 127-node-labyrinth with a water port at the terminal, Rosenberg et al. (2021) observed that the navigation behavior of water-restricted mice is influenced not solely by the extrinsic water reward but also by intrinsic motivators, including their curiosity to explore the environment.

Inverse reinforcement learning (IRL) (Ng et al., 2000; Arora & Doshi, 2021) is a popular approach to recover a reward function that induces the observed behavior, assuming that the demonstrator was

(softly) maximizing its long-term return. Along with the significant successes of IRL in autonomous driving (Kalweit et al., 2020; Nasernejad et al., 2023), robotics (Kumar et al., 2023; Chen et al., 2023), and healthcare domains (Coronato et al., 2020; Chan & van der Schaar, 2021), it appears to be emerging as an valuable tool for constructing mathematical behavior models in neuroscience research, as exemplified by Yamaguchi et al. (2018), Kwon et al. (2020), and Alyahyay et al. (2023). Classic IRL methods seek to identify a single, fixed reward function for a specific scenario. In contrast, Ashwood et al. (2022a) suggested that animal's goals can evolve over time due to factors like fatigue, satiation, and curiosity. Under this assumption, they proposed the Dynamic Inverse Reinforcement Learning (DIRL) framework, which parametrizes the animal's reward function as a time-varying linear combination of a small number of spatial reward maps, which are referred to as "goal maps". By positing the existence of multiple goal maps with time-varying weights, DIRL allows the instantaneous reward function to vary *continuously* in time. This innovative framework achieved state-of-the-art performance in characterizing animal behavior. Nevertheless, persistent demands have emerged regarding an IRL framework incorporating *discrete* time-varying reward functions, particularly following the proposal by Ashwood et al. (2022b) that natural behaviors can be represented through a Markov chain characterized by alternating between discrete intentions.

To address this requirement, we propose the novel class of *Latent (Markov) Variable Inverse Q-learning (L(M)V-IQL)* algorithms, which extend the single reward Inverse Q-learning (IQL) framework from Kalweit et al. (2020) to solve IRL problems accounting for multiple intentions. We formulate an Expectation-Maximization (EM) approach to first cluster animal trajectories into multiple intentions, and then solving the IRL problem independently for each intention, respectively. We theoretically demonstrate that L(M)V-IQL can cover the most common two types of intention transition dynamics: generalized Bernoulli process and Markov process. Finally, we present the application of our framework in 1) a simulated Gridworld environment, 2) real mice navigation trajectories with known environment model from Rosenberg et al. (2021), and 3) real mice decision-making data from a dynamic two-armed bandit task with unknown environment model, showing that our methods outperform state-of-the-art in behavior prediction, demonstrate exceptional aptitude in capturing the intentions of animals, and provide interpretable reward functions corresponding to each identified intention.

## 2 RELATED WORK

Table 1: Overview of different multiple intention IRL algorithms.

| Algorithms | Model-free | Rewards | # Intentions | Time-varying Rewards |
|---|---|---|---|---|
| EM-MLIRL (Babes et al., 2011) | × | linear | known | × |
| DPM-BIRL (Choi & Kim, 2012) | × | linear | unknown | × |
| MRP/MPO-MC (Dimitrakakis & Rothkopf, 2012) | × | linear | known | discrete |
| BN-IRL (Michini & How, 2012) | × | linear | unknown | discrete |
| BNP-IRL (Surana & Srivastava, 2014) | × | linear | unknown | discrete |
| Meta-AIRL (Gleave & Habryka, 2018) | ✓ | non-linear | × | × |
| SEM/MCEM-MIIRL (Bighashdel et al., 2021) | × | non-linear | unknown | × |
| MI-$\Sigma$-GIRL (Likmeta et al., 2021) | ✓ | linear | known | × |
| DIRL (Ashwood et al., 2022a) | × | non-linear | known | continuous |
| **L(M)V-IAVI (Ours)** | × | non-linear | known | discrete |
| **L(M)V-IQL (Ours)** | ✓ | non-linear | known | discrete |

Various approaches have been introduced to address multiple intention inverse reinforcement learning problems (Table 1). Notably, several frameworks based on parametric (Babes et al., 2011; Likmeta et al., 2021), or Bayesian non-parametric (Choi & Kim, 2012; Bighashdel et al., 2021) approaches share a similar objective to ours, allowing for multiple agents with distinct reward functions. However, these frameworks do not accommodate single agents with time-varying rewards. Gleave & Habryka (2018) developed a meta adversarial learning method for multi-task IRL problems. While their framework demonstrated high-level performance in real-world applications, it sacrifices theoretical interpretability and heavily relies on exploiting similarities between reward functions across tasks. Built on the Bayesian IRL approach, algorithms from Dimitrakakis & Rothkopf

(2012), Michini & How (2012), and Surana & Srivastava (2014) posits that an agent's trajectory can be divided into discrete behavioral states with corresponding unique reward function. Nevertheless all these algorithms assume linear reward functions and the Bayesian inference problem is intractable even for moderately sized finite-state IRL problems. Finally, the framework from Ashwood et al. (2022a) parametrizes the animal's reward function as a time-varying linear combination of a small number of spatial reward maps with Gaussian random walk prior over weights, capturing continuous time-varying reward functions. Their approach pursue a related aim to ours, yet is limited to capturing continuous intra-episode variation of reward functions (as illustrated in Section 5.1), and difficult to adopt to other environments. Last but not least, most of the aforementioned algorithms are model-based, relying on a known transition dynamics of the environment, whereas in many scenarios, the environment model is unknown.

## 3 BACKGROUND

### 3.1 INVERSE REINFORCEMENT LEARNING

Consider a Markov Decision Process (MDP): $\{\mathcal{S}, \mathcal{A}, T, \mathcal{R}, \gamma\}$, where $\mathcal{S}$ and $\mathcal{A}$ denotes the state- and action-space; $T\colon \mathcal{S} \times \mathcal{A} \to \Delta(\mathcal{S})$ is the state transition function ($\Delta$ denotes the probability simplex) with $T(s, a, s') := \Pr(s' \mid s, a)$; $\mathcal{R}\colon \mathcal{S} \times \mathcal{A} \to \mathbb{R}$ defines the reward function, and $\gamma \in [0, 1)$ denotes the discount factor. The problem of inverse reinforcement learning is formally defined as:

**Problem 1** (IRL problem). *Given the demonstration space $\mathcal{D} := \{\xi_i\}_{i=1}^{N}$ with $N$ trajectories, where each trajectory is a sequence of state-action pairs, $\xi_i := \{(s_1, a_1), (s_2, a_2), \ldots\}$, the IRL problem consists of finding a reward function $\mathcal{R}$ that maximizes the LL between agent demonstrations and the (soft) optimal policy $\pi_{\mathcal{R}}$ under $\mathcal{R}$:*

$$\underset{\mathcal{R}}{\text{maximize}} \quad \sum_{i=1}^{N} \log(\Pr(\xi_i \mid \pi_{\mathcal{R}})). \tag{1}$$

### 3.2 INVERSE Q-LEARNING

The class of Inverse Q-learning algorithms (Kalweit et al., 2020) provides a precise yet notably time-efficient solution to Problem 1, compared to the popular Maximum Entropy IRL algorithm from Ziebart et al. (2008) and some of its variants. It assumes that the demonstrations are collected from an agent following a Boltzmann policy according to its unknown optimal value function $Q^*$:

$$\pi^E(s, a) := \frac{\exp(Q^*(s, a))}{\sum_{A \in \mathcal{A}} \exp(Q^*(s, A))} \Rightarrow Q^*(s, a) = Q^*(s, b) + \log(\pi^E(s, a)) - \log(\pi^E(s, b)), \tag{2}$$

for all actions $a \in \mathcal{A}$ and $b \in \mathcal{A}_{\bar{a}}$ where $\mathcal{A}_{\bar{a}} := \mathcal{A} \backslash \{a\}$. Using the Bellman optimality equation in Equation 2, the immediate reward of action $a$ in state $s$ can be expressed by the immediate reward of some other action $b \in \mathcal{A}_{\bar{a}}$, the respective log-probabilities and future action-values:

$$r(s, a) = \eta_s^a + \frac{1}{d_{\mathcal{A}} - 1} \sum_{b \in \mathcal{A}_{\bar{a}}} \left[ r(s, b) - \eta_s^b \right], \tag{3}$$

where $r(s, a) \in \mathcal{R}$ is the unknown reward function, $d_{\mathcal{A}}$ denotes the dimension of $\mathcal{A}$, and $\eta_s^a := \log(\pi^E(s, a)) - \gamma \sum_{s' \in \mathcal{S}} T(s, a, s') \max_{a' \in \mathcal{A}} Q^*(s', a')$. The resulting system of linear equations can be solved with least squares, leading to the model-based Inverse Action-value Iteration (IAVI) algorithm, which solves the IRL problem analytically in *closed-form*. To relax the assumption of an existing transition model and action probabilities, IAVI was further extended to the sampling-based model-free Inverse Q-learning (IQL) algorithm (Kalweit et al., 2020). They showed that the Boltzmann distribution induced by the optimal action-value function on the learned reward from IAVI and IQL is equivalent to the arbitrary demonstrated behavior distribution.

## 4 INVERSE Q-LEARNING ABOUT MULTIPLE INTENTIONS

We first define the Multiple Intention Inverse Reinforcement Learning (MI-IRL) problem accordingly:

**Problem 2** (MI-IRL problem). *Let $\mathcal{Z} := \{z_k\}_{k=1}^K$ be a $K$-dimensional latent state space with each $z_k \in \mathcal{Z}$ corresponding to one intention, and let $\mathcal{D} := \{\xi_i\}_{i=1}^N$ be $N$ trajectories demonstrated by an agent each under one of the latent states without labels. The MI-IRL problem consists of inferring the latent state labels and the corresponding reward functions $\{\mathcal{R}_k\}_{k=1}^K$ in $\mathcal{D}$ such that under the $k$-th latent state the agent (softly) optimizes $\mathcal{R}_k$.*

We adopt the EM (Dempster et al., 1977) as a straightforward approach to attack Problem 2. Let $\Theta$ be the set of parameters to be inferred, and let $\mathcal{Y} := \{y_i\}_{i=1}^N$ be the set of latent state labels for each trajectory $\xi \in \mathcal{D}$, where $y_i = k$ if trajectory $i$ came from under latent state $z_k$. At iteration $\tau$ of the EM process before $\Theta$ converges, the expected value of the likelihood function $\mathcal{L}$ of $\Theta$ will be maximized as in the following update equation:

$$\Theta^{\tau+1} := \arg\max_{\Theta} \sum_{\mathcal{Y}} \mathcal{L}(\Theta \mid \mathcal{D}, \mathcal{Y}) \Pr(\mathcal{Y} \mid \mathcal{D}, \Theta^{\tau}) \tag{4}$$

Noting that different latent state transition dynamics lead to respective parameter space $\Theta$ and specific implementations of Equation 4. In the following, we consider the latent state transition dynamics described with a generalized Bernoulli process (independent latent states) and a Markov process (Markovian interdependent latent states).

### 4.1 CLUSTERING OF INDEPENDENT LATENT STATES

We start from the simpler case where the occurrence of different intentions satisfies a generalized Bernoulli process. Let $\{\nu_1, \ldots, \nu_K \mid \nu_1 + \cdots + \nu_K = 1\}$ be the set of prior probability corresponding to the occurrence of each latent state $z_k$, the set of parameters $\Theta$ to be inferred is then $\{\nu_1, \ldots, \nu_K; \mathcal{R}_1, \ldots, \mathcal{R}_K \mid \nu_1 + \cdots + \nu_K = 1\}$. The optimal value for respective parameters in this parameter set at each EM iteration is provided by Theorem 1:

**Theorem 1.** *Given that the intention transition dynamics satisfies a generalized Bernoulli process, at iteration $\tau$, the EM update equation (Equation 4) for each parameter in the corresponding parameter set $\Theta = \{\nu_1, \ldots, \nu_K; \mathcal{R}_1, \ldots, \mathcal{R}_K \mid \nu_1 + \cdots + \nu_K = 1\}$ is given by*

$$\begin{cases} \nu_k^{\tau+1} := \dfrac{1}{N} \sum_{i=1}^N \zeta_{ik}^{\tau} \\[2mm] \mathcal{R}_k^{\tau+1} := \arg\max_{\mathcal{R}_k} \sum_{i=1}^N \zeta_{ik}^{\tau} \log(\Pr(\xi_i \mid \pi_{\mathcal{R}_k})), \end{cases} \tag{5}$$

*for all $\nu, \mathcal{R} \in \Theta$, where $\zeta_{ik} := \frac{1}{Z} \nu_k \prod_{(s,a) \in \xi_i} \pi_{\mathcal{R}_k}(s, a)$ is the probability that trajectory $i$ was demonstrated under latent state $z_k$ normalized by factor $Z$.*

*Proof.* See Appendix A.1. □

Noting that updating the reward function estimation $\mathcal{R}$ at each iteration according to Equation 5 is equivalent to solving Problem 1, except that each trajectory is weighted by a probability $\zeta$ during sampling. Thus combining the above EM approach for trajectory clustering with IAVI or IQL algorithms leads to the class of Latent Variable Inverse Q-learning (LV-IQL) algorithms (cf. Algorithm 1), solving the MI-IQL problem (Problem 2) when the occurrence of different latent states is independent.

## 4.2 CLUSTERING OF MARKOVIAN INTERDEPENDENT LATENT STATES

In addition to the generalized Bernoulli process, the Markov process is also considered an alternative for describing intention transition dynamics (Ashwood et al., 2022b; Le et al., 2023), where the occurrence of the next latent state corresponding to a specific intention is dependent and only dependent on the previous latent state. Under this assumption, given the agent demonstrations $\mathcal{D}$ consisting of a sequence of trajectories, the set of parameters to be inferred is then $\{\Pi, \Lambda; \mathcal{R}_1, \dots, \mathcal{R}_K\}$, where $\Pi \colon \Delta(\mathcal{Z})$ and $\Lambda \colon \mathcal{Z} \to \Delta(\mathcal{Z})$ ($\Delta$ is the probability simplex) denoting the latent state initial distribution probability and latent state transition matrix, respectively. The optimal value for respective parameters in the parameter set $\Theta$ at each EM iteration is provided by Theorem 2:

**Theorem 2.** *Given that the intention transition dynamics satisfies a Markov process, at iteration $\tau$, the EM update equation (Equation 4) for each parameter in the corresponding parameter set $\Theta = \{\Pi, \Lambda; \mathcal{R}_1, \dots, \mathcal{R}_K\}$ is given by*[1]

$$
\begin{cases}
\pi_k^{\tau+1} := \Pr(y_0 = k \mid \mathcal{D}, \Theta^\tau) \\[2mm]
\Lambda_{kl}^{\tau+1} := \dfrac{\sum_{i=1}^{N} \Pr(y_{i-1} = k, y_i = l \mid \mathcal{D}, \Theta^\tau)}{\sum_{i=1}^{N} \Pr(y_{i-1} = k \mid \mathcal{D}, \Theta^\tau)} \\[4mm]
\mathcal{R}_k^{\tau+1} := \arg\max_{\mathcal{R}_k} \sum_{i=0}^{N} \Pr(y_i = k \mid \mathcal{D}, \Theta^\tau) \log(\Pr(\xi_i \mid \pi_{\mathcal{R}_k})),
\end{cases}
\tag{6}
$$

*for all $\pi, \Lambda, \mathcal{R} \in \Theta$.*

*Proof.* See Appendix A.2. ∎

In practice, the Forward-Backward algorithm (Baum et al., 1970) can be used to address the probabilities in Equation 6, and IAVI or IQL then estimates the corresponding reward function independently for each latent state. This leads to the class of Latent Markov Variable Inverse Q-learning (LMV-IQL) algorithms (cf. Algorithm 2, details see also Appendix B.2), which solves Problem 2 when the latent state transition satisfies the Markov property.

## 5 EXPERIMENTS

### 5.1 APPLICATION OF LV-IQL TO SIMULATED BEHAVIOR

We first demonstrate the LV-IQL algorithm on trajectories from a simulated animal foraging task in a $15 \times 15$ Gridworld environment (Figure 1), and compare to the class of single intention IQL algorithms. The action space of Gridworld was defined as $\mathcal{A} := \{up, down, left, right, stay\}$. Stochastic transitions took the agent in a random direction with 30% chance after each action execution. Two types of rewarded resources were randomly assigned to each state in the environment. The agent was considered to have two intentions: 'Hungry' and 'Thirsty' with the occurrence probability of 70% and 30% respectively. Under the 'hungry' intention, states with food resource assigned would be rewarded $(+1)$ while states with water resource would be punished $(-1)$, and vice versa under the 'thirsty' intention. Each trajectory was demonstrated under one of the two intentions with the agent executing the optimal greedy policy on the respective reward function (Figure 2, Top, Ground Truth). (More details see also Appendix C.1.)

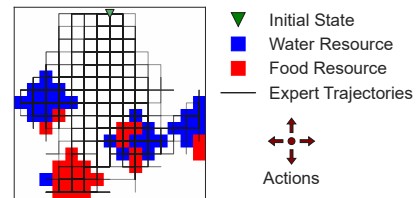

Figure 1: Gridworld environment architecture with agent trajectories.

We compared between the performance of LV-IAVI, LV-IQL, IAVI, and IQL trained on the whole demonstration space. Two latent states were considered for both LV-IAVI and LV-IQL. As a measure

---

[1] Here we assume the index of trajectories in $\mathcal{D}$ starts from 0 instead of 1 for convenience but without losing generality.

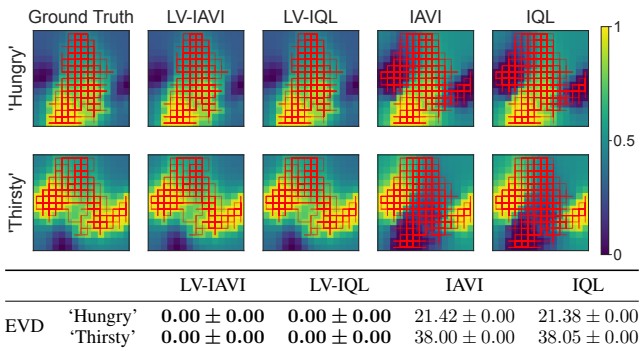

| | | LV-IAVI | LV-IQL | IAVI | IQL |
|---|---|---|---|---|---|
| EVD | 'Hungry' | **0.00 ± 0.00** | **0.00 ± 0.00** | 21.42 ± 0.00 | 21.38 ± 0.00 |
| | 'Thirsty' | **0.00 ± 0.00** | **0.00 ± 0.00** | 38.00 ± 0.00 | 38.05 ± 0.00 |

Figure 2: **(Top)** Visualization of the normalized ground truth and learnt state-value functions. Red lines indicate the ground truth trajectory distribution and the learnt trajectory clusters used to recover the reward function for respective intention. (All expert trajectories are shown for each figure of IAVI and IQL.) **(Bottom)** EVD for different approaches, Mean ± SE over 5 repeated runs.

of performance, we used the Expected Value Difference (EVD) metric (Levine et al., 2011). EVD is defined as the mean square error between the state-value under the true reward function for the expert policy and the state-value under the true reward for the optimal Boltzmann policy w.r.t. the learnt reward. It provides an estimation of the sub-optimality of the learnt policy under the true reward function. For LV-IAVI and LV-IQL, the inferred latent states with respective trajectory clusters were assigned to the best-fit ground truth intentions. Since IAVI and IQL assumed all trajectories were demonstrated under one intention, the EVD was analyzed twice on the ground truth reward for different intentions with the same learnt Boltzmann policy (Figure 2). The trajectory clusters learnt with LV-IAVI and LV-IQL are highly overlapped with the ground truth trajectory distribution. As a result, the learnt reward functions via LV-IAVI and LV-IQL match the respective ground truth reward functions exactly, while the single intention IAVI and IQL only resulted in a large EVD of $\sim 21$ for the 'hungry' intention and $\sim 38$ for the 'thirsty' intention, representing a mixed reward function for the two intentions. Similar results were found when we removed the punishment on intention irrelevant rewards (Appendix C.2). Noting that the DIRL algorithm (Ashwood et al., 2022a) is infeasible here as it assumes continuously time-varying rewards, which only addresses cases where the intention transition occurs after each action execution. However, in the above Gridworld experiment, each episode was conducted under one of two intentions, where the intention remains fixed within the episode, and the transition between intentions only occurs between episodes.

## 5.2 APPLICATION OF LMV-IAVI TO MICE NAVIGATING TRAJECTORIES

Next, we apply the LMV-IAVI algorithm to mice trajectories recorded during navigating in a 127-node labyrinth environment (Rosenberg et al., 2021) (Figure 3A) as a benchmark to compare with the state-of-the-art — DIRL (Ashwood et al., 2022a). In this task, two groups of mice navigated a labyrinth: one with water restrictions and access to a water port (Figure 3A), and another without water restrictions and no access to water. (More details see also Appendix D.1.) To formalize the MDP, we consider a 127 state environment with known world model and action space $\mathcal{A} := \{left, right, reverse, stay\}$.

We demonstrate model comparison by first applying our method to trajectories from the water-restricted animals. The test set log-likelihood (LL) is similar for LMV-IAVI and DIRL under single intention ($K = 1$). However, LMV-IAVI with $K > 2$ substantially outperforms DIRL (Figure 3B). Although the test LL continues to grow for larger $K$, the Bayesian information criterion (BIC) appears to increase (Figure 3C). Thus LMV-IAVI with 2 latent states is considered for subsequent analysis. The learnt mice policy under latent state 1 ('Tired') displays a preference of moving out from the water port towards the maze entrance and stay, while the policy under latent state 2 ('Thirsty') guides the mice directly to the water port along the optimal track. Correspondingly, in the 'Tired' latent state, the highest state occupancy is noted at the entrance state, while under 'Thirsty', it is observed at the water port (Figure 3D). To delve into the intention transition dynamics, we computed the posterior probability over mice's latent state across all trajectories. The recovered average tem-

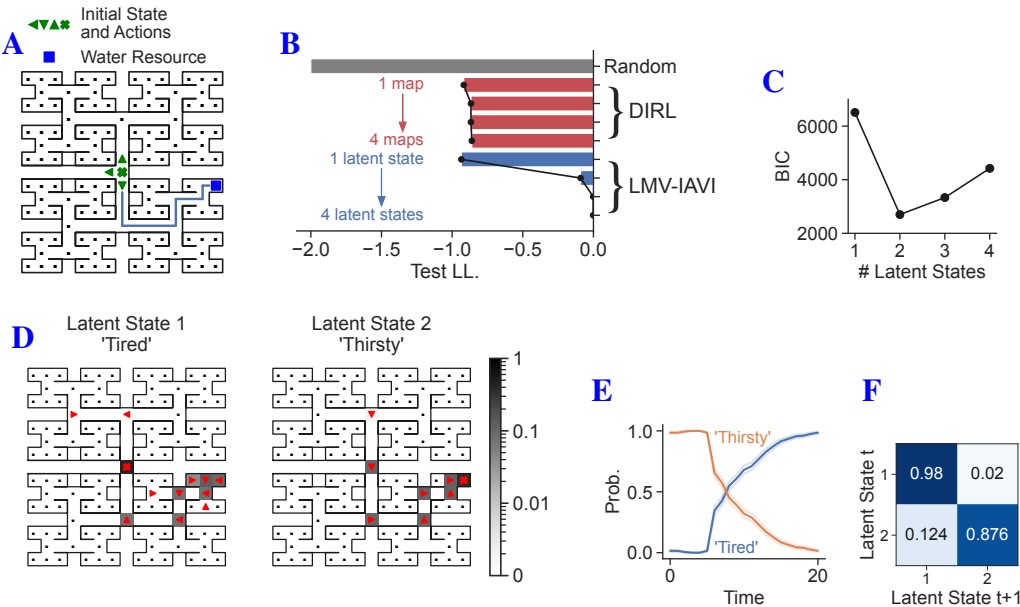

Figure 3: **(A)** The labyrinth environment. Blue line shows the optimal path from entrance to water port. *Left*, *right*, and *reverse* actions are represented with arrows while *stay* is denoted with cross. **(B)** Comparison of LMV-IAVI on test set trajectories to a random policy and DIRL, represented as LL. **(C)** BIC as a function of latent state numbers in LMV-IAVI. **(D)** Learnt policy (red arrows and crosses) in the environment and corresponding state occupancy (grey colormap) under different intentions. State occupancy was calculated by assigning each trajectory to the latent state with highest posterior probability. Policies are shown only for states with non-zero occupancy. **(E)** Trajectories of latent state probabilities. Solid and shaded curves denote the Mean and SE. **(F)** Inferred latent state transition matrix from the best-fitting LMV-IAVI.

poral latent state trajectories show a high probability of the 'Thirsty' latent state at the beginning but later on tailed off, as the 'Tired' latent state gradually became dominant (Figure 3E). These findings demonstrate that LMV-IAVI not only excels the state-of-the-art in predicting mice labyrinth navigating behavior, but also provides distinct and interpretable reward functions. Similarly, our LMV-IAVI algorithm again outperforms DIRL when applied to the water-unrestricted animal trajectory dataset. Further details can be found at Appendix D.2.

## 5.3 APPLICATION OF LMV-IQL TO MICE REVERSAL-LEARNING BEHAVIOR

Finally, we apply the LMV-IQL algorithm to behavioral data recorded from a group of mice engaged in a dynamic two-armed bandit reversal-learning task from De La Crompe et al. (2023). At the beginning of the task, water-restricted mice may choose from two available spouts, left (L) and right (R), with random one of them assigned water as extrinsic reward. After reaching an online performance of 75% correct in a 15-trials sliding average window and a minimum 20-trials block, the rewarded spout is automatically changed. To formulate the MDP, we define the action space as: $\mathcal{A} := \{left, right\}$. Every state $s \in \mathcal{S}$ is defined with a set of truncated history information: $s_t := \{\varphi_{t-1}, \ldots, \varphi_{t-\ell_h}; a_{t-1}, \ldots, a_{t-\ell_h}\}$, where $\ell_h$ denotes the history length, $a \in \mathcal{A}$ denotes history action, and $\varphi \in \{correct, error\}$ represents history environmental feedback, i.e. extrinsic reward. Such MDP formulation allows us to avoid explicitly describing a partially observable MDP formulation. Different from the first two experiments, the environment model here is considered to be unknown in the dynamic reversal-learning task.

We begin our application of LMV-IQL on the recorded mice behavior by selecting the hyper-parameter $\ell_h$. At this step, we only consider single latent state LMV-IQL (equivalent to IQL). We compared the LL on training and test sets of multiple IQL fitting with different $\ell_h$ (Figure 4A).

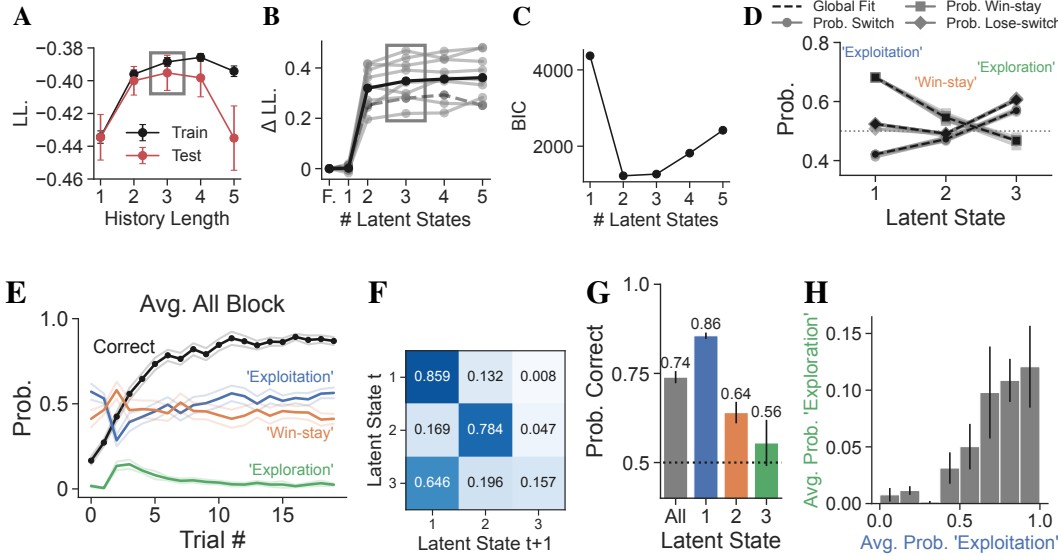

Figure 4: **(A)** LL (Mean ± SE, 5-fold cross-validation) as a function of $\ell_h$ of single latent state LMV-IQL. **(B)** Change in test LL as a function of latent state numbers in LMV-IQL with $\ell_h = 3$, relative to the fQ-learning model (labeled 'F.'). Each trace represents a single mouse, averaged over cross-validation. Solid black indicates the mean across animals, and the dashed curve indicates the example mouse. **(C)** BIC as a function of latent state numbers in LMV-IQL with $\ell_h = 3$. **(D)** Learnt mice policy represented with the probability of switch, win-stay, and lose-switch. Each grey curve denotes one mouse. **(E)** Average task performance and trajectories of latent state probabilities. Solid and shaded curves denote the Mean and SE. **(F)** Inferred latent state transition matrix from the best-fitting LMV-IQL for the example mouse. **(G)** Overall task performance (gray) and the performance under different latent states. **(H)** Relationship between the probability of the 'Exploitation' latent state, 5 trials before block switch and the mean probability of the 'Exploration' latent state, 5 trials after block switch, Mean ± SE.

The LL on test sets shows a bell-shaped curve as $\ell_h$ increases, indicating an overfit on the training set when $\ell_h > 3$. Noting that there is an abnormal drop on training set LL at large $\ell_h$s. This can be explained with the insufficient sampling given the fixed set of expert demonstrations, since the size of the state space $\mathcal{S}$ grows exponentially as the history length $\ell_h$ increases. The best test LL is achieved at $\ell_h = 3$, which is selected for subsequent steps. Next, to determine the number of intentions $K$ under which mice demonstrated the trajectories, we fit multiple LMV-IQL with varying numbers of latent states. In this step, we additionally applied a forgetting Q-learning (fQ-learning) model (Beron et al., 2022), which has been widely recognized as a prominent forward behavioral model for the reversal-learning task. This was done using the same dataset, serving as a baseline for comparative analysis. We found that the multiple intention LMV-IQL fitting substantially outperformed the single intention models (Figure 4B). Although the BIC w.r.t. different $K$ indicates that both $K = 2$ and $K = 3$ are reasonable values (Figure 4C), we will focus subsequent analysis on the LMV-IQL with 3 latent states for biological interpretability. (More details about LMV-IQL fitting see also Appendix E.)

The inferred mice policies from LMV-IQL define how the subjects make decisions under three intentions (Figure 4D). One of these policies, operating within latent state 1, displays a strong inclination toward adopting a 'win-stay' and 'lose-switch' strategy, which is the optimal policy in this deterministic reward bandit task. On the other hand, within latent state 2, the policy, referred to as the 'Win-stay' policy, exhibits a preference for exploitation when the previous trial was successful. However, following error trials, it employs a random action selection strategy, indicated by a $\sim 0.5$ probability of executing a 'lose-switch'. Lastly, in latent state 3, a characteristic 'Exploration' policy emerges, where the subject consistently favors selecting the option opposite to the one chosen in the preceding trial, irrespective of whether they had won or lost

in that particular instance. The recovered latent state trajectories in the example session reveals that the most probable latent state often exhibits a probability close to 1, indicating a high degree of confidence in discerning the subject's intent based on the observed data (Figure 5). The 'Exploration' intention predominantly manifested at the onset of a block and endured for a relatively brief duration, in alignment with the learned latent state transition matrix (Figure 4F).

The significant values along the diagonal of the transition matrix within latent states 1 and 2, corresponding to 'Exploitation' and 'Win-stay', signify a heightened preference for persisting in the same latent state over multiple consecutive trials. Additionally, it becomes evident that error trials tend to coincide with the trials where the posterior probability of 'Win-stay' and 'Exploration' latent states reaches its zenith, corroborating the presence of suboptimal exploratory behavior associated with these two intentions. The average latent state transition trajectories across all blocks closely resembles those observed in the example session (Figure 4E). As each block begin with the animal's performance at a relatively low level, there is a decline in the posterior probability associated with the

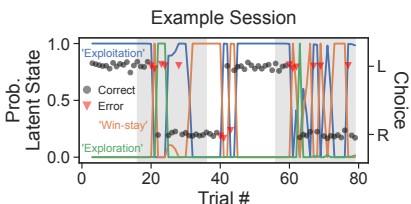

Figure 5: Posterior latent state probabilities for an example session. Dots and triangles indicate mice behavior.

'Exploitation' latent state, accompanied by an increase in the probabilities of the other two latent states associated with suboptimal exploratory strategies. Nonetheless, as the subjects' performance steadily improves, the 'Exploitation' latent state progressively reasserts its dominance. Finally, to quantify latent state occupancies across all sessions, we assigned each trial to its most probable state. In contrast to the cohort's general correct rate of $0.74 \pm 0.02$, mice performed significantly better within the 'Exploitation' latent state, achieving a correctness rate of $0.86 \pm 0.01$. In comparison, they attained lower correctness rates of $0.64 \pm 0.03$ and $0.56 \pm 0.06$ in the two alternative latent states (Figure 4G). Furthermore, it's worth noting that the mean posterior probability of the 'Exploration' latent state at the beginning of a new block shows a positive correlation with the average probability of the 'Exploitation' state at the end of the preceding block (Figure 4H), suggesting that the 'Exploration' latent state appears to involve a deliberate, exploration-oriented action selection when mice are highly engaged and possess a good understanding of the environment.

## 6 CONCLUSION

In this study, we introduce a novel class of *Latent (Markov) Variable Inverse Q-learning (L(M)V-IQL)* algorithms for characterizing animal behavior during complex decision-making tasks. We extend the class of IQL algorithms (Kalweit et al., 2020) to learn multiple discrete reward functions from demonstrations. Specifically, we address the two most prevalent types of intention transition dynamics: the generalized Bernoulli process and the Markov process, under both model-based and model-free contexts. To validate our framework and compare with the state-of-the-art, we conduct experiments on simulated and real animal behavior data. Our approaches demonstrate a substantial improvement in behavior prediction compared to DIRL (Ashwood et al., 2022a) on mice navigation trajectories (indicated by the LL on held-out trajectories), without losing interpretability of the learnt reward functions. Moreover, our method provides distinct and interpretable reward functions for the mice cohort engaged in the reversal-learning task, where the animals displayed a pattern of alternating between exploitation and exploration intentions, which could extend over several consecutive trials within a single session. The transitions between these intentions followed a typical block-correlated trajectory, wherein the mice were more likely to exhibit in exploratory behaviors at the start of a new block, particularly if they had been highly engaged in the task in the preceding block.

A compelling avenue for future research lies in extending our framework to involve function approximations, which would enable the learning of a low-dimensional embedding of each state in the environment via e.g. a deep neural network. Such extension would allow us to scale our approach to high-dimensional or continuous state spaces, while also enabling the generalization across states. Another promising direction would be to extend the fixed intention transition probabilities with e.g. a generalized linear model, to incorporate the identification of potential external factors that influence intention transition dynamics.

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

APPENDICES

# A  PROOF OF THEOREMS

## A.1  PROOF OF THEOREM 1

*Proof.* Substitute the parameters $\Theta = \{\nu_1, \ldots, \nu_K; \mathcal{R}_1, \ldots, \mathcal{R}_K \mid \nu_1 + \cdots + \nu_K = 1\}$ under independent latent state assumption into the EM update equation (Equation 4) and unroll:

$$\Psi(\Theta, \Theta^\tau) \coloneqq \sum_{\mathcal{Y}} \mathcal{L}(\Theta \mid \mathcal{D}, \mathcal{Y}) \Pr(\mathcal{Y} \mid \mathcal{D}, \Theta^\tau) \qquad \text{(from Equation 4)}$$

$$= \sum_{\mathcal{Y}} \sum_{i=1}^{N} \log(\nu_{y_i} \Pr(\xi_i \mid \pi_{\mathcal{R}_{y_i}})) \prod_{i'=1}^{N} \Pr(y_{i'} \mid \xi_{i'}, \Theta^\tau) \qquad (\text{A.1})$$

$$= \sum_{y_1} \cdots \sum_{y_N} \sum_{i=1}^{N} \sum_{k=1}^{K} \delta_{k=y_i} \log(\nu_k \Pr(\xi_i \mid \pi_{\mathcal{R}_k})) \prod_{i'=1}^{N} \Pr(y_{i'} \mid \xi_{i'}, \Theta^\tau) \qquad (\text{A.2})$$

$$= \sum_{k=1}^{K} \sum_{i=1}^{N} \log(\nu_k \Pr(\xi_i \mid \pi_{\mathcal{R}_k})) \sum_{y_1} \cdots \sum_{y_N} \delta_{k=y_i} \prod_{i'=1}^{N} \Pr(y_{i'} \mid \xi_{i'}, \Theta^\tau) \qquad (\text{A.3})$$

$$= \sum_{k=1}^{K} \sum_{i=1}^{N} \log(\nu_k \Pr(\xi_i \mid \pi_{\mathcal{R}_k})) \zeta_{ik}^\tau \qquad \text{(by Equation 5)}$$

$$= \sum_{k=1}^{K} \sum_{i=1}^{N} \zeta_{ik}^\tau \log(\nu_k) + \sum_{k=1}^{K} \sum_{i=1}^{N} \zeta_{ik}^\tau \log(\Pr(\xi_i \mid \pi_{\mathcal{R}_k})), \qquad (\text{A.4})$$

where $\delta$ denotes the Kronecker delta function. Equation A.4 indicates that $\nu_k$ and $\mathcal{R}_k$ are not inter-dependent, we can thus optimize them separately in the M-step of EM, leading to the second update equation in Equation 5 trivially. According to Gibbs' inequality, the first term of Equation A.4 is maximized if and only if

$$\nu_k^{\tau+1} \coloneqq \frac{1}{N} \sum_{i=1}^{N} \zeta_{ik}^\tau, \qquad (\text{A.5})$$

for all $\nu \in \Theta$, proving the first update equation in Equation 5. $\qquad \square$

## A.2  PROOF OF THEOREM 2

*Proof.* Similar to the proof for Theorem 1, substitute the parameter set $\Theta = \{\Pi, \Lambda; \mathcal{R}_1, \ldots, \mathcal{R}_K\}$ into the EM update equation (Equation 4) and unroll:

$$\Psi(\Theta, \Theta^\tau) \coloneqq \sum_{\mathcal{Y}} \mathcal{L}(\Theta \mid \mathcal{D}, \mathcal{Y}) \Pr(\mathcal{Y} \mid \mathcal{D}, \Theta^\tau) \qquad \text{(from Equation 4)}$$

$$= \sum_{\mathcal{Y}} \log(\pi_{y_0} \Pr(\xi_0 \mid \pi_{\mathcal{R}_{y_0}}) \prod_{i=1}^{N} \Lambda_{y_{i-1} y_i} \Pr(\xi_i \mid \pi_{\mathcal{R}_{y_i}})) \Pr(\mathcal{Y} \mid \mathcal{D}, \Theta^\tau) \qquad (\text{A.6})$$

$$= \sum_{\mathcal{Y}} \log(\pi_{y_0}) \Pr(\mathcal{Y} \mid \mathcal{D}, \Theta^\tau)$$

$$+ \sum_{\mathcal{Y}} \sum_{i=1}^{N} \log(\Lambda_{y_{i-1} y_i}) \Pr(\mathcal{Y} \mid \mathcal{D}, \Theta^\tau)$$

$$+ \sum_{\mathcal{Y}} \sum_{i=0}^{N} \log(\Pr(\xi_i \mid \pi_{\mathcal{R}_{y_i}})) \Pr(\mathcal{Y} \mid \mathcal{D}, \Theta^\tau) \qquad (\text{A.7})$$

---

**Algorithm 1:** Latent Variable Inverse Q-learning (LV-IQL)

**Input:** agent demonstrations $\mathcal{D}$, latent space dimension $K$

1   initialize $\Theta := \{\nu_1, \ldots, \nu_K; \mathcal{R}_1, \ldots, \mathcal{R}_K \mid \nu_1 + \cdots + \nu_K = 1\}$

2   **repeat**

3      **E-step**

4         **foreach** $\xi_i \in \mathcal{D}$ **do**

5            **forall** $k$ **do**

6               $\zeta_{ik} \leftarrow \prod_{(s,a)\in\xi_i} \pi_{\mathcal{R}_k}(s,a)\nu_k / Z$

7      **M-step**

8         **forall** $k$ **do**

9            $\nu_k \leftarrow \sum_{i=1}^{N} \zeta_{ik}/N$

10           compute $\mathcal{R}_k$ via IAVI or IQL on $\mathcal{D}$ with weight $\zeta_{ik}$ on trajectory $\xi_i$

11   **until** convergence;

    **Output:** $\Theta$

---

$$
\begin{aligned}
&= \sum_{y_1} \cdots \sum_{y_N} \sum_{k=1}^{K} \delta_{k=y_0} \log(\pi_k) \prod_{i'=1}^{N} \Pr(y_{i'} \mid \xi_{i'}, \Theta^\tau) \\
&\quad + \sum_{y_1} \cdots \sum_{y_N} \sum_{i=1}^{N} \sum_{k=1}^{K} \sum_{l=1}^{K} \delta_{k=y_{i-1}, l=y_i} \log(\Lambda_{kl}) \prod_{i'=1}^{N} \Pr(y_{i'} \mid \xi_{i'}, \Theta^\tau) \\
&\quad + \sum_{y_1} \cdots \sum_{y_N} \sum_{i=0}^{N} \sum_{k=1}^{K} \delta_{k=y_i} \log(\Pr(\xi_i \mid \pi_{\mathcal{R}_k})) \prod_{i'=1}^{N} \Pr(y_{i'} \mid \xi_{i'}, \Theta^\tau) \quad\quad \text{(A.8)} \\
&= \sum_{k=1}^{K} \Pr(y_0 = k \mid \mathcal{D}, \Theta^\tau) \log(\pi_k) \\
&\quad + \sum_{k=1}^{K} \sum_{l=1}^{K} \sum_{i=1}^{N} \Pr(y_{i-1} = k, y_i = l \mid \mathcal{D}, \Theta^\tau) \log(\Lambda_{kl}) \\
&\quad + \sum_{k=1}^{K} \sum_{i=0}^{N} \Pr(y_i = k \mid \mathcal{D}, \Theta^\tau) \log(\Pr(\xi_i \mid \pi_{\mathcal{R}_k})), \quad\quad \text{(A.9)}
\end{aligned}
$$

where $\delta$ denotes the Kronecker delta function. Since $\pi_k$, $\Lambda_{kl}$ and $\mathcal{R}_k$ are not interdependent, we can thus maximize the respective term separately, resulting in Equation 6. $\qquad\square$

**Remark 1.** *In a more practical case where the agent demonstration space has multiple trajectory sequences, Theorem 2 can also be generalized and proved in the same manner.*

**Remark 2.** *All $\xi \in \mathcal{D}$ above are assumed to be the trajectory for a whole episode. In some special cases where it is assumed that the latent state transition happens after each action execution, instead of per episode, Theorem 2 can also be applied by regarding each episode as a trajectory sequence with each trajectory consists of only one action execution.*

## B   ALGORITHMS

### B.1   LATENT VARIABLE INVERSE Q-LEARNING

The pseudo code for LV-IQL can be found at Algorithm 1.

## B.2 Latent Markov Variable Inverse Q-learning

To implement the LMV-IQL algorithm, let the forward probability $\mathfrak{a}_{ik}$ be the posterior probability of the observed agent demonstrations up until trajectory $i$ and the latent state under which the $i$-th trajectory was demonstrated is $z_k$:

$$
\begin{aligned}
\mathfrak{a}_{ik} &:= \Pr(\mathcal{D}_{0:i}, z_i = k \mid \Theta) \\
&= \begin{cases}
\Pi_k \Pr(\xi_0 \mid z_0 = k, \Theta), & i = 0 \\
\sum_{j=1}^{K} \mathfrak{a}_{(i-1)j} \Lambda_{jk} \Pr(\xi_i \mid z_i = k, \Theta), & i \neq 0,
\end{cases}
\end{aligned}
\tag{B.1}
$$

and the backward probability $\mathfrak{b}_{ik}$ be the posterior probability of the demonstrations after trajectory $i$:

$$
\begin{aligned}
\mathfrak{b}_{ik} &:= \Pr(\mathcal{D}_{i+1:N} \mid z_i = k, \Theta) \\
&= \begin{cases}
\sum_{j=1}^{K} \mathfrak{b}_{(i+1)j} \Lambda_{kj} \Pr(\xi_{i+1} \mid z_{i+1} = j, \Theta), & i \neq N \\
1, & i = N.
\end{cases}
\end{aligned}
\tag{B.2}
$$

The posterior probability that trajectory $i$ was demonstrated under latent state $z_k$ is then denoted as:

$$
\begin{aligned}
\mathfrak{g}_{ik} &:= \Pr(y_i = k \mid \mathcal{D}, \Theta) \\
&= \frac{\mathfrak{a}_{ik} \mathfrak{b}_{ik}}{\sum_{j=1}^{K} \mathfrak{a}_{ij} \mathfrak{b}_{ij}},
\end{aligned}
\tag{B.3}
$$

and the posterior probability that trajectory $i - 1$ was demonstrated under latent state $z_k$ and concomitantly trajectory $i$ was demonstrated under latent state $z_l$ is:

$$
\begin{aligned}
\mathfrak{x}_{ikl} &:= \Pr(y_{i-1} = k, y_i = l \mid \mathcal{D}, \Theta) \\
&= \frac{\mathfrak{a}_{(i-1)k} \Lambda_{kl} \Pr(\xi_i \mid z_i = l, \Theta) \mathfrak{b}_{il}}{\sum_{u=1}^{K} \sum_{v=1}^{K} \mathfrak{a}_{(i-1)u} \Lambda_{uv} \Pr(\xi_i \mid z_i = v, \Theta) \mathfrak{b}_{iv}}.
\end{aligned}
\tag{B.4}
$$

Thus the update equation in Equation 6 is equivalent to

$$
\begin{cases}
\Pi_k^{\tau+1} := \mathfrak{g}_{0k}^{\tau} \\
\Lambda_{kl}^{\tau+1} := \dfrac{\sum_{i=1}^{N} \mathfrak{x}_{ikl}^{\tau}}{\sum_{i=0}^{N-1} \mathfrak{g}_{ik}^{\tau}} \\
\mathcal{R}_k^{\tau+1} := \arg\max_{\mathcal{R}_k} \sum_{i=0}^{N} \mathfrak{g}_{ik}^{\tau} \log(\Pr(\xi_i \mid \Pi_{\mathcal{R}_k})).
\end{cases}
\tag{B.5}
$$

Combining Equation B.5 and the class of IQL algorithms leads to LMV-IQL (Algorithm 2).

## C    Further Details and Additional Results on the Simulated Gridworld Behavior Dataset

### C.1    The Gridworld Dataset and Model Training

The simulated agent demonstration space from the Gridworld environment consisted of 512 trajectories with each having a length of 64 movements. The discount factor was set to be $\gamma = 0.99$. All evaluated algorithms were trained for 5 repeated runs on the whole demonstration space until

---

**Algorithm 2:** Latent Markov Variable Inverse Q-learning (LMV-IQL)

---

**Input:** expert demonstrations $\mathcal{D}$, latent space dimension $K$

1  initialize $\Theta := \{\Pi, \Lambda; \mathcal{R}_1, \ldots, \mathcal{R}_K\}$
2  **repeat**
3      **E-step**
4          calculate $\mathfrak{g}$ and $\mathfrak{x}$ according to Equations B.1–B.4
5      **M-step**
6          **forall** $k$ **do**
7              $\Pi_k \leftarrow \mathfrak{g}_{0k}$
8              **forall** $l$ **do**
9                  $\Lambda_{kl} \leftarrow \sum_{i=1}^{N} \mathfrak{x}_{ikl} / \sum_{i=0}^{N-1} \mathfrak{g}_{ik}$
10             compute $\mathcal{R}_k$ via IAVI or IQL on $\mathcal{D}$ with weight $\mathfrak{g}_{ik}$ on trajectory $\xi_i$
11 **until** convergence;
    **Output:** $\Theta$

---

convergence (difference of learnt reward function and the posterior probability of intentions for each trajectory $< 10^{-3}$ between iterations).

## C.2   Additional Results on Gridworld

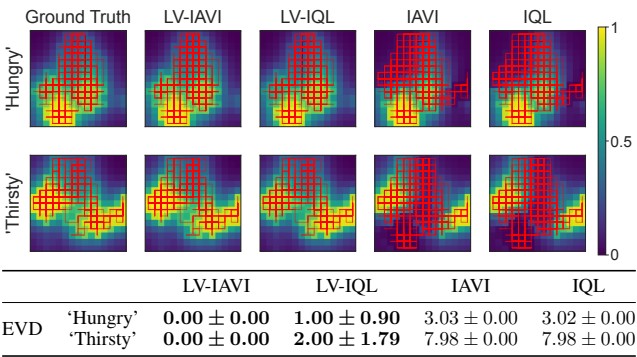

| | | LV-IAVI | LV-IQL | IAVI | IQL |
|---|---|---|---|---|---|
| EVD | 'Hungry' | $\mathbf{0.00 \pm 0.00}$ | $\mathbf{1.00 \pm 0.90}$ | $3.03 \pm 0.00$ | $3.02 \pm 0.00$ |
| | 'Thirsty' | $\mathbf{0.00 \pm 0.00}$ | $\mathbf{2.00 \pm 1.79}$ | $7.98 \pm 0.00$ | $7.98 \pm 0.00$ |

Figure 6: **(Top)** Visualization of the normalized ground truth and learnt state-value functions. Red lines indicate the ground truth trajectory distribution and the learnt trajectory clusters used to recover the reward function for respective intention. (All expert trajectories are shown for each figure of IAVI and IQL.) **(Bottom)** EVD for different approaches, Mean $\pm$ SE over 5 repeated runs.

We also performed analysis under the environment set up where the intention irrelevant punishments were removed, i.e. replacing the $-1$ reward on the type of reward irrelevant to the intentions with 0. In this environment, there is an increased overlapping between some of the demonstrated trajectories under different intentions (Figure 6). However, LV-IAVI and LV-IQL still outperform the single intention algorithm IAVI and IQL in trajectory clustering and recovering corresponding expert reward functions.

## D   Further Details and Additional Results on the Evaluation of Mice Navigation Trajectories

### D.1   Labyrinth Navigation Task and Model Training

In the navigation task from Rosenberg et al. (2021), two cohorts of 10 mice moved freely in dark through the labyrinth over the course of 7 hours. For comparability with the result from Ashwood et al. (2022a), we obtained their pre-processed mouse trajectories for water-restricted and

water-unrestricted animals from `https://github.com/97aditi/dynamic_irl`[2]. For the pre-processing, Ashwood et al. (2022a) used a clustering algorithm (based on DBSCAN (Ester et al., 1996)) for aligning trajectories across animals and bouts to reduce variability. After the pre-processing, they obtained 200 trajectories from the water-restricted animals and 207 trajectories from the water-unrestricted animals. 20% of trajectories from each cohort were held out as a test set.

To compare the performance of LMV-IAVI and DIRL in this environment, we used the source code provided by Ashwood et al. (2022a) to train DIRL on the animal trajectory dataset. All LMV-IAVI algorithms were trained for 10 repeated runs with different initializations, and the results from the initializations with hightest test set LL was selected for analysis. The initial latent state distribution $\Pi$ was initialized with a uniform distribution on the latent state space $\mathcal{Z}$ as $\Pi := \mathcal{U}(\mathcal{Z})$, and the latent state transition matrix $\Lambda$ was initialized as: $\Lambda := 0.95 \times I + \mathcal{N}(0, 0.05 \times I)$, where $\mathcal{N}$ denotes the normal distribution and $I : \mathcal{Z} \times \mathcal{Z} \to \mathbb{R}$ is the identity matrix. This initial $\Lambda$ was then normalized so that each row added up to 1. The discount factor was set to be $\gamma = 0.99$.

## D.2 ADDITIONAL RESULTS FOR WATER-UNRESTRICTED MICE

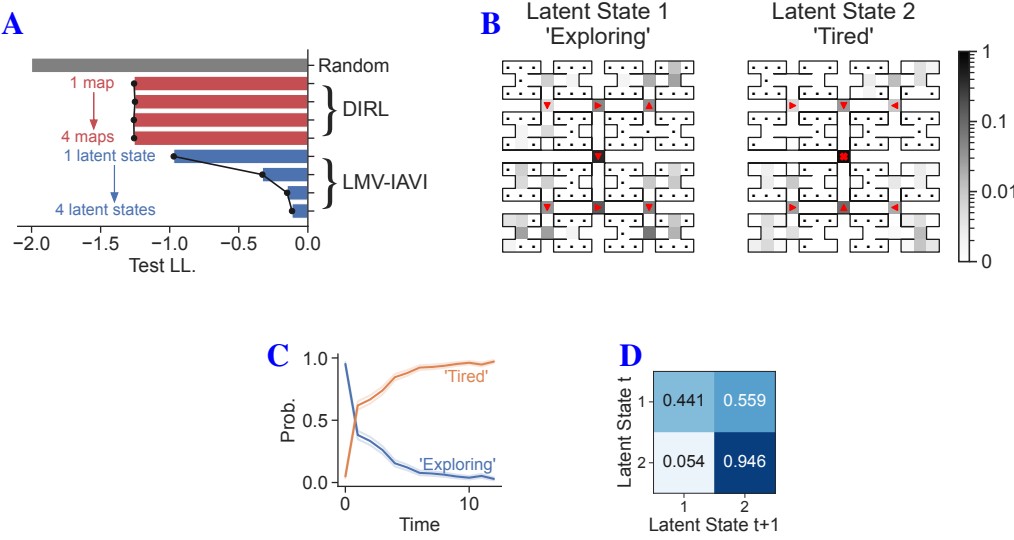

Figure 7: **(A)** Comparison of LMV-IAVI on test set trajectories to a random policy and DIRL, represented as LL. **(B)** Learnt policy (red arrows) under different intentions and corresponding state occupancy (grey colormap) in the environment. State occupancy was calculated by assigning each trajectory to the latent state with highest posterior probability. Policies are shown only for some states. **(C)** Trajectories of latent state probabilities. olid and shaded curves denote the Mean and SE. **(D)** Inferred latent state transition matrix from the best-fitting LMV-IAVI.

In contrast to the outcomes from the water-restricted animal dataset (Figure 3), LMV-IAVI demonstrates a higher test LL, even when considering a single intention. As the number of latent states (associated with DIRL's goal maps) grows, the test LL of LMV-IAVI increases, while the test LL of DIRL remains constant (Figure 7A). Focus on the two latent states LMV-IAVI, the inferred policy under two intentions exhibits 'Exploring' and 'Tired' behavior. The policy under 'Exploring' tends to encourage the animal lingering in the labyrinth, whereas the policy under 'Tired' latent state steers the animal back to the maze entrance. Correspondingly, the posterior probability of 'Exploring' initially dominates at the session's beginning but is generally surpassed by the 'Tired' latent state over time.

---

[2]The original recorded animal trajectories from Rosenberg et al. (2021) are provided with MIT open source license at `https://github.com/markusmeister/Rosenberg-2021-Repository`.

# E    FURTHER DETAILS ON THE EVALUATION OF MICE REVERSAL-LEARNING BEHAVIOR

The behavior data was collected from a cohort of mice consisted of 9 mice in total. Behavior recordings for each mice were repeated for at least 7 independent sessions with an average of $\sim 87$ trials per session.

We employed a multi-stage fitting procedure (Algorithm 3) to select hyper-parameters and to allow us to fit LMV-IQL individually to each animal. In the first stage, we concatenated the data from all animals in a single dataset together. We then performed multiple IQL (single latent state LMV-IQL) with different history truncation length $\ell_h \in \{1, \ldots, 5\}$ on the concatenated data. Out of the 5 different values, We chose the $\ell_h$ that resulted in the best test set LL for subsequent stages. In the second stage, we run multiple LMV-IQL with different number of latent states $K \in \{2, \ldots, 5\}$ again to the concatenated dataset to obtain a global fit. The initial latent state distribution $\Pi$ was initialized with a uniform distribution on the latent state space $\mathcal{Z}$ as $\Pi := \mathcal{U}(\mathcal{Z})$, and the latent state transition matrix $\Lambda$ was initialized as: $\Lambda := 0.95 \times I + \mathcal{N}(0, 0.05 \times I)$, where $\mathcal{N}$ denotes the normal distribution and $I \colon \mathcal{Z} \times \mathcal{Z} \to \mathbb{R}$ is the identity matrix. This initial $\Lambda$ was then normalized so that each row added up to 1. The reward and action-value function was initialized as $r(s, a) := \mathcal{N}(0, 0.2)$ and $Q(s, a) := \mathcal{N}(0, 5)$ for all $s \in \mathcal{S}$ and $a \in \mathcal{A}$. All discount factors were set to be $\gamma = 0.99$. Since Algorithm 2 is not guaranteed to converge to the global optimum (Salakhutdinov et al., 2003), we performed 10 different initializations for each value of $K$. Out of the 10 initializations, we chose the parameters that resulted in the best training set LL for subsequent stages. In the last stage of the fitting procedure, we wanted to obtain a respective but aligned LMV-IQL fit for each animal, so we initialized the parameters for each animal with the best global fit parameters from all animals together, omitting the necessity to permute the retrieved latent states from each animal so as to map semantically similar intentions to one another. Algorithm 3 shows the pseudo-code for the whole procedure. A 5-fold cross-validation was used to split the training and test dataset[3], and Algorithm 3 was fit on each cross-validation fold independently.

---

**Algorithm 3:** Fitting LMV-IQL on real mice behavior

```
Fit IQL globally:
```
1  **foreach** $\ell_h \in \{1, \ldots, 5\}$ **do**
2      run IQL on the concatenated data from all animals until convergence
3  select best $\ell_h$ with largest test set LL
```
Fit LMV-IQL globally:
```
4  **foreach** $K \in \{2, \ldots, 5\}$ **do**
5      **foreach** $i \in \{1, \ldots, 10\}$ **do**
6          initialize LMV-IQL with $K$ latent states and random parameters
7          run Algorithm 2 on the concatenated data from all animals until convergence

```
Fit separate LMV-IQL to each animal:
```
8  **forall** animals **do**
9      **foreach** $K \in \{1, \ldots, 5\}$ **do**
10         initialize LMV-IQL with $K$ latent states using the best global fit parameters for this $K$
11         run Algorithm 2 until convergence

---

[3]Here we considered to hold out entire sessions of behavior for assessing test set performance. That is, the training and test set consisted of 80% and 20% of recorded sessions of each mouse, respectively.

