# OpenReview forum: "L(M)V-IQL: Multiple Intention Inverse Reinforcement Learning for Animal Behavior Characterization"
_ICLR.cc/2024/Conference — Submitted to ICLR 2024_

### Official Review · Reviewer_yMx7 · 2023-10-27

**Soundness:** 3 good
**Presentation:** 3 good
**Contribution:** 2 fair
**Rating:** 6
**Confidence:** 4

**Summary:**

In this work, the Authors propose a novel algorithm for discerning and characterizing multiple intents underlying natural behaviors. To this end, they combine the inverse Q-learning (a version of IRL) with the expectation-maximization algorithm used to delineate the intents. The Authors consider two types of dynamics in their framework: the Bernoulli process and the Markov process; they provide theoretical derivations and describe implementations for both. They then test their framework on a simulated task (a gridworld with 2 underlying intentions) where they show the framework’s ability to recover the ground truth, and on an existing mouse dataset (a bandit reversal learning task) where they reconstruct and describe mouse intents.

**Strengths:**

A definite strength of this work is that it has been anticipated in the field. As a continuous-time model for multiple intent reconstruction has been developed by Ashwood et al (NeurIPS 2022; referenced in current manuscript), a question that has been repeatedly raised was about the discrete version of the framework. This question has become especially relevant after another work by Ashwood et al (Nat Neurosci 2022; also referenced in current manuscript) that used large-scale IBL data to propose that natural behaviors can be represented via an MDP featuring rapid progression of discrete states. This ICLR submission delivers on that expectation.

The model in the paper is well-founded; it features reasonable choices of the constituting algorithms (e.g., the Baum-Welch algorithm for discerning the intents).

Having the analysis for both simulated and animal data is a plus.

**Weaknesses:**

Along the same line with the strengths, I see the main weakness here in high similarity to Ashwood et al (NeurIPS 2022) work. The Authors mention in the Appendix that the aforementioned approach “is limited to capturing continuous intra-episode variation of reward functions during navigation behavior, and difficult to adopt to other environments” but, should that be true, that requires further substantiation. The data analyses offered in this paper seem to mainly serve as a proof of principle for the proposed model.

Overall, the work is nicely done and well-timed; my only concern is that the high similarity to prior literature determines the work’s novelty which may be insufficient for ICLR. With that said, I’m open to comments by the Authors, other Reviewers, and the Area Chair in that regard.

**Questions:**

-Introduction: wouldn’t it make more sense to introduce your work via Ashwood et al, 2022a and 2022b papers? I feel like this way the reason for the development of your model and the comparison to the existing state-of-the-art would be more transparent.

-Page 5 under Equation 10: what does Delta Z mean? Is it supposed to reflect the available transitions?

-Page 6 under Figure 3: why is it necessary to punish the types of reward irrelevant to the intentions? A more natural way seemingly would be to set them equal to zero. I assume this natural way hasn’t worked out for some reason?

-Figure 4A: why does the LL in the training curve drop? That is unlikely to be explained by overfitting as suggested in the text.

-Page 7 bottom line: “Although model performance continued to improve slightly with more latent states, we will focus […] on […] 3 states”. Wouldn’t it be easier to make this argument by using the Bayesian Information Criterion instead of the pure NLL to choose K? This way one can arrive at a principled number of intents that very well may turn out to be equal to 2.

-Figure 5C. Following up on my previous point, this figure leaves me with the impression that the third intent is just spurious (not stable; immediately reverses to the first intent). Could you please comment on why you consider it important?

Minor comments:

-Figure 3: the overlap of the crosses (x) and dots ( . ) is hard to follow. Could you please use an alternative way to represent this data?

I’d also suggest tuning down a couple of literature-related claims:

-Page 2: “[IRL’s] adoption as mathematical behavior models in neuroscience research has been relatively limited”. I had another impression – it seems to be an up-and-coming tool, as exemplified by some awesome works from Jon Pillow’s and Xaq Pitkow’s groups.

-Page 2: “[our method presents a […] framework for characterizing the delicate balance between exploration and exploitation […] which constitutes a […] comparatively understudied aspect within the realm of neuroscience.” I’d say that, first, there’s a huge spillover from the machine learning field of intrinsic motivation (a.k.a. an internal reward for exploration); many of these works claim biological plausibility. There are some other nice works, e.g. Pisupati et al (eLife 2021) and references therein that directly address the issue. There’s also lots of work on Bayesian optimality that study the deviations from optimal exploitation to account for the environmental dynamics, e.g. Yu and Cohen (NeurIPS 2008).

-Page 4: “In behavioral neuroscience, it is commonly considered that animals alternate between multiple intentions under the Markov property”. The entire reason why the Ashwood et al (Nat Neurosci 2022) paper cited there emerged is because that’s _not_ how people used to characterize natural behaviors. This is reflected literally in the first sentence of the said paper. While this new work has gotten substantial traction in the field, I wouldn’t say that that new way to model data has completely wiped out the conventional approach.

________________________________________________________________________________
post-rebuttal:

I would like to thank the Authors for their clarifications. I appreciated the fast, detailed responses.
Posting my final response here as, at this time, I cannot otherwise make it visible to the Authors.

I believe that the updated manuscript is a more solid, transparent, and substantiated work.
The most interesting finding to me is that new priors here allowing for abrupt changes of goal maps, enabled by the novel problem formulation, optimization objective, and solver, were more consistent with the mouse decision-making data in the maze experiment than DIRL (the previous SOTA), rendering the proposed model important. I would also like to thank the Authors for the clarification that the choice of different smoothness prior in DIRL would not necessarily be able to recover the same dynamics, necessitating the formulation of the problem in the way proposed here.

Despite the similarity to prior work, this is an important and interesting result. I increased my score to reflect it.

---

> ### Author Response · Authors · 2023-11-22
>
> We thank reviewer yMx7 for his suggestions to improve our submission. We address your concerns in detail below and have updated our paper accordingly.
>
> *”The Authors mention in the Appendix that the aforementioned approach “is limited to capturing continuous intra-episode variation of reward functions during navigation behavior, and difficult to adopt to other environments” but, should that be true, that requires further substantiation.”*
>
> Thanks for your question. The DIRL approach from Ashwood et al. (NeurIPS 2022) assumes the rewards vary continuously over time, i.e. the intention transition happens after each action execution. The limitation of their approach can be indicated from the Gridworld environment in our first experiment, where we consider each episode was demonstrated under one of two intentions, i.e. the intention is fixed within each episode, and the intention transition only happens between episodes. DIRL will not be feasible under such environments since the variation of reward functions is, however, discrete. In contrast, our method is viable in cases where the transition of intention occurs either after each action execution, i.e. continuous (as discussed in Remark 2 in Appendix B.2), or at the end of each episode, i.e. discrete (as theoretically demonstrated in Appendix B.2). We have elucidated this argument in the current revision to provide a clearer understanding of the adaptability and applicability of our approach in diverse settings.
>
> *”Introduction: wouldn’t it make more sense to introduce your work via Ashwood et al, 2022a and 2022b papers? I feel like this way the reason for the development of your model and the comparison to the existing state-of-the-art would be more transparent.”*
>
> Thank you for your valuable suggestion! We have revised the introduction to transparently articulate our motivation for this work.
>
> *”Page 5 under Equation 10: what does Delta Z mean? Is it supposed to reflect the available transitions?”*
>
> Thanks for your question. $\Delta(\mathcal{Z})$ indicates the probability simplex on the latent state space. Thus $\Pi \colon \Delta(\mathcal{Z})$ defines the latent state initial distribution probability, and $\Lambda \colon \mathcal{Z} \mapsto \Delta(\mathcal{Z})$ reflects the latent state transition matrix, i.e. the latent state distribution probability at $t+1$ given the latent state at $t$. We have refined the definition accordingly in the current revision to eliminate any ambiguity.
>
> *”Page 6 under Figure 3: why is it necessary to punish the types of reward irrelevant to the intentions? A more natural way seemingly would be to set them equal to zero. I assume this natural way hasn’t worked out for some reason?”*
>
> Thank you for your suggestion. We have conducted additional experiments on the Gridworld environment with the same resource distribution but removed the punishment on the intention irrelevant reward, i.e. replacing the -1 reward on the type of reward irrelevant to the intentions with 0. Related results are shown in Appendix C.2. Our algorithms still worked under this configuration. However, in this case there was an increased overlapping between some of the demonstrated trajectories under different intentions (Figure 6, Top), resulting in a slight decrease of the performance of LV-IQL, but LV-IAVI and LV-IQL still outperformed the single intention algorithm IAVI and IQL in trajectory clustering and recovering corresponding expert reward functions.
>
> *”Figure 4A: why does the LL in the training curve drop? That is unlikely to be explained by overfitting as suggested in the text.”*
>
> Thanks for your question. As we consider each state $s_t$ in the state space $\mathcal{S}$ to be defined with a set of truncated history information to avoid explicitly describing a partially observable MDP formulation for the reversal-learning task, the size of the state space $\mathcal{S}$ will grow exponentially as the history truncation length $\ell_h$ increases. Thus the sampling on some states given the fixed animal demonstrations could be insufficient for the IQL to reconstruct a precise reward function via stochastic approximation. The abnormal drop of the training LL will be omitted given a larger animal demonstration dataset. We have updated this explanation in the current revision for better clarification.

---

> > ### Author Response · Authors · 2023-11-22
> >
> > *”Page 7 bottom line: “Although model performance continued to improve slightly with more latent states, we will focus […] on […] 3 states”. Wouldn’t it be easier to make this argument by using the Bayesian Information Criterion instead of the pure NLL to choose K? This way one can arrive at a principled number of intents that very well may turn out to be equal to 2.”*
> > *”Figure 5C. Following up on my previous point, this figure leaves me with the impression that the third intent is just spurious (not stable; immediately reverses to the first intent). Could you please comment on why you consider it important?”*
> >
> > Thanks for your question. We have calculated the BIC for LMV-IQL across a varying number of latent states in the current revision (Figure 4C). It is shown that the BIC for the number of latent states $K = 2$ and $K = 3$ are similar, suggesting that both value are reasonable choices for $K$. The reason that the 3 latent states model was preferred was that the 3 latent states model provided a fine grained information about exploratory behavior, i.e. the disengaged, passive exploration (‘Win-stay’, performing random action selection after lose trails) and the active exploration (‘Exploration’, choosing the other side compared to the preceding trial regardless of win or lose) (Figure 4D). Figure 4H also shows that the mean posterior probability of the ‘Exploration’ latent state during the initial 5 trials of a new block showed a positive correlation with the average probability of the ‘Exploitation’ state during the final 5 trials of the preceding block, further supporting this point that the ‘Exploration’ latent state appears to involve a deliberate, exploration-oriented action selection only when mice are highly engaged and possess a good understanding of the environment. The ‘Exploration’ latent state gives the opportunity to characterize a type of learning strategy based on error. Since it has been shown that rodents (Kononowicz et al. PNAS 2022), as well as humans, can learn from their errors, we then think that maintaining the exploration latent state confers a clear advantage in terms of biological interpretability. Additionally, the instability of the ‘Exploration’ latent state as observed here could be due to our two-armed bandit reversal-learning task design. We considered a performance-determined reward switch here, i.e. the rewarded spout would be changed only when the subject reached an online performance of 75% correct in a 15-trials sliding average window and a minimum of 20-trials block. Thus the high fraction of ‘Exploration’ latents states would lead to unfinished sessions (each session was required to have 4 blocks), which was filtered out from our dataset.
> >
> > *”Figure 3: the overlap of the crosses (x) and dots ( . ) is hard to follow. Could you please use an alternative way to represent this data?”*
> >
> > Thanks, we have updated the data representation in the current revision. Instead of depicting a t-SNE visualization of expert trajectories (Figure 3 from the prior revision), the pertinent results are now depicted in Figure 2 (Top), where trajectories are directly plotted within the Gridworld environment as red lines, with the corresponding state-values displayed in the background.
> >
> > *”Page 2: “[IRL’s] adoption as mathematical behavior models in neuroscience research has been relatively limited”. I had another impression – it seems to be an up-and-coming tool, as exemplified by some awesome works from Jon Pillow’s and Xaq Pitkow’s groups.”*
> > *”Page 2: “[our method presents a […] framework for characterizing the delicate balance between exploration and exploitation […] which constitutes a […] comparatively understudied aspect within the realm of neuroscience.” I’d say that, first, there’s a huge spillover from the machine learning field of intrinsic motivation (a.k.a. an internal reward for exploration); many of these works claim biological plausibility. There are some other nice works, e.g. Pisupati et al (eLife 2021) and references therein that directly address the issue. There’s also lots of work on Bayesian optimality that study the deviations from optimal exploitation to account for the environmental dynamics, e.g. Yu and Cohen (NeurIPS 2008).”*
> > *”Page 4: “In behavioral neuroscience, it is commonly considered that animals alternate between multiple intentions under the Markov property”. The entire reason why the Ashwood et al (Nat Neurosci 2022) paper cited there emerged is because that’s not how people used to characterize natural behaviors. This is reflected literally in the first sentence of the said paper. While this new work has gotten substantial traction in the field, I wouldn’t say that that new way to model data has completely wiped out the conventional approach.”*
> >
> > Thanks for your suggestions. We have updated these statements.
> >
> > Does this address your concerns?

---

> > > ### Comment · Reviewer_yMx7 · 2023-11-22
> > >
> > > Thanks for your detailed response.
> > >
> > > I've read the response, the communication with other Reviewers, and the updated text. With this, my technical questions have been fully addressed. I appreciate the clarity of the updated manuscript and the added experiments and comparisons. I am specifically excited about the new comparison with DIRL, although I may need some extra time to process the significance of this result (e.g., whether there are circumstances under which DIRL would reach similar performance and, related, whether either of the two methods in comparison can be viewed as a limit case of another one).
> > >
> > > As it stands, this manuscript is now a complete work. The only concern that remains and that has been raised by most of the Reviewers is the degree of similarity to prior work and whether the offered novelty matches the expectations for the ICLR. I look forward to discussing that matter with the other Reviewers and with the Area Chair.

---

> ### Comment · Reviewer_yMx7 · 2023-11-23
> **Follow-up questions**
>
> After carefully examining the added experiment (the comparison to DIRL), I have a few follow-up questions.
>
> In Figure 3F of the updated manuscript, the transition probability is ~2% from "thirsty" to "tired" and ~1% from "tired" to thirsty" suggesting that a transition would likely happen at most once per session and the state of the animal throughout the session would be thus described by a theta (step) function. Figure 3E, to my understanding, is smooth and not step-like due to averaging over sessions/mice. The better performance of the new model compared to DIRL would then be explained by the underlying truth (the satiation in mice) being more consistent with a step function rather than with a smoothly decaying function (consistent with a few previous works). Therefore, my questions are:
>
> -would DIRL be able to reconstruct the same result if larger changes in goal maps are allowed? In DIRL, if I recall it correctly, smoothness in the goal map weights is imposed via a Gaussian prior which, to my understanding, could be varied.
>
> -would the ability to vary the smoothness prior in DIRL render the current work as a limit case of DIRL?
>
> -In one of the replies here, the Authors state: *"An essential characteristic of our approach is that we allow both latent state transition within an episode (i.e. continuous-time transition similar to DIRL, as shown in the second and third experiment) and between episodes (i.e. stay constant within one episode, as shown in the first experiment)."* Could you please clarify what is meant by the continuous-time transition in the proposed method? Is this in the sense that, on different rollouts, the switch between the states would occur at different times, thus making the transition "smooth" on average, or does it mean something different?
>
> Thanks in advance.

---

> > ### Author Response · Authors · 2023-11-23
> >
> > Thanks for your follow-up question. We will address them in detail below.
> >
> > *”would DIRL be able to reconstruct the same result if larger changes in goal maps are allowed? In DIRL, if I recall it correctly, smoothness in the goal map weights is imposed via a Gaussian prior which, to my understanding, could be varied.”*
> > *”would the ability to vary the smoothness prior in DIRL render the current work as a limit case of DIRL?”*
> >
> > To the best of our understanding, the IRL problem is an ill-posed problem since in theory there are an infinite number of reward functions (a.k.a. solutions) maximizing the probability of observing the expert demonstrations. If we recall it correctly, the Gaussian prior on time-varying weights $\mathbf{\alpha}$ of DIRL also plays the role as a constraint of the weights in order that the problem would then be tractable. This constraint is evident in their objective function, wherein $\mathbf{\alpha}$ is optimized using gradient descent (refer to Equation 6 of Ashwood et al. (NeurIPS, 2022)). Although theoretically one can vary different priors to change the smoothness of the goal map transition, the objective function and optimization solver, however, should also be modified carefully correspondingly to keep the IRL problem solvable. Consequently, we posit that exploring different priors of DIRL may not be a trivial endeavor without further substantiation, a consideration that could potentially extend beyond the scope originally outlined in the DIRL paper (assuming the configuration in Ashwood et al. (NeurIPS, 2022) is already optimized).
> >
> > On the other hand, we propose that the continuous time-varying reward functions of DIRL could be approximated by L(M)V-IQL by increasing the number of latent states (an extreme case would be to have the same number of latent states as the length of the time horizon), at the cost of increased computation time. The underlying goal maps of DIRL can then be reconstructed via e.g. clustering methods on the inferred latent states from L(M)V-IQL. We believe that this avenue merits consideration for future investigations.
> >
> > Lastly, a noteworthy contribution of our approach lies in the facilitation of ***model-free*** learning. In stochastic environments where the environment model (a.k.a. state transition function of the MDP) is normally considered to be unknown (such as the reversal-learning task in our third experiment), the model-based DIRL remains infeasible. This stems from the requisite dependency on the environment model for the calculation of their action-value function and policy. In contrast, L(M)V-IQL demonstrates the capability to conduct model-free inference through stochastic approximation on the action-value functions. We assert that this feature holds significant importance, particularly in real-world environments and the evolving landscape of behavioral experiments, where stochasticity assumes a pivotal role.
> >
> > In a word, although there are similar characteristics between L(M)V-IQL and DIRL, we propose that they are not equivalent, as a result of the problem formulation, optimization objective and solver, and feasibility across different environments. Certainly, the comparison between our approach and DIRL presents an interesting question that would lead to valuable future work. We welcome any additional insights or perspectives you may offer on this matter and remain open to further discourse.
> >
> > *”In one of the replies here, the Authors state: "An essential characteristic of our approach is that we allow both latent state transition within an episode (i.e. continuous-time transition similar to DIRL, as shown in the second and third experiment) and between episodes (i.e. stay constant within one episode, as shown in the first experiment)." Could you please clarify what is meant by the continuous-time transition in the proposed method? Is this in the sense that, on different rollouts, the switch between the states would occur at different times, thus making the transition "smooth" on average, or does it mean something different?”*
> >
> > Allow us to provide further clarification on the term "continuous-time transition" in the context of L(M)V-IQL. In L(M)V-IQL, this denotes the "continuous time transition of discrete intentions", wherein the term "continuous-time" specifies that the transition takes place after each action selection. It is essential to note that this is the smallest time interval from the observer's perspective, despite remaining discrete in nature. This is similar to the GLM-HMM model from Ashwood et al. (Nat. Neuroscience, 2022). Importantly, this does not imply that, across different rollouts, the switch between states occurs at varying times. On the contrary, given a specific dataset, the global optimum of L(M)V-IQL fitting remains fixed, resulting in a consistent timing of intention transition.
> >
> > Does this clarify your question?

---

### Official Review · Reviewer_o7yT · 2023-10-30

**Soundness:** 3 good
**Presentation:** 3 good
**Contribution:** 2 fair
**Rating:** 5
**Confidence:** 4

**Summary:**

This work considers an inverse reinforcement learning model (IRL) with latent discrete intention variables. Using an inverse Q-learning and EM-based approach, they perform inference on these latent variables at each time based on either generalized Bernoulli or Markovian dynamics, learning both the transition dynamics between intentions (in the Markov case) and their corresponding reward functions. Experiments involve recovery on simulated data and from behavior in mice performing a two-alternative forced choice task with randomly changing reward structure.

**Strengths:**

- Intriguing generalization of inverse RL methods to neuroscience.
- Well-motivated incorporation of latent drives.

**Weaknesses:**

- The fitted model is somewhat simplistic. Latent states are assumed to be multinomial or Markov, but the most plausible biological assumption would be that transitions between drives also interact with reward/satiety/recent history.
- There are only two experiments: one on simulated data (where it is) compared to IAVI and IQL but not the Ashwood et al. or other similar models that might be applicable. Similarly, the mouse behavior is quite limited in terms of the need for RL. Again, the Ashwood Nature Neuro paper or the Ebitz, Albarran, and Moore (2018) provide fairly flexible models that are likely to capture the data as well. Given the synthetic data, one would have expected a more challenging task here as a target for IRL.

**Questions:**

- Is it possible to incorporate some recent reward history into the transition structure? Since the inference algorithm is EM, will any EM-compatible latent variable model work, in principle?
- Where are the bottlenecks for the method in terms of inferential complexity? Is the limiting factor the IAVI regression (i.e., the size of the tabular problem) or the EM complexity?

---

> ### Author Response · Authors · 2023-11-22
>
> We thank reviewer o7yT for his suggestions to improve our submission. We address your concerns in detail below and have updated our paper accordingly.
>
> *”There are only two experiments: one on simulated data (where it is) compared to IAVI and IQL but not the Ashwood et al. or other similar models that might be applicable. Similarly, the mouse behavior is quite limited in terms of the need for RL. Again, the Ashwood Nature Neuro paper or the Ebitz, Albarran, and Moore (2018) provide fairly flexible models that are likely to capture the data as well. Given the synthetic data, one would have expected a more challenging task here as a target for IRL.”*
>
> Thanks for your great suggestion. We have included an additional experiment on the application of LMV-IAVI to real mice navigation trajectories within a labyrinth from Rosenberg et al. (2021). This dataset was also considered as a benchmark for IRL algorithm comparison in Ashwood et al. (NeurIPS 2022), since it would be challenging to define a suitable reward function to apply forward models (such as the GLM-HMM model from Ashwood et al. (Nat. Neuroscience, 2022) and the framework from Ebitz et al. (2018)). In this experiment, we compared our approaches with the state-of-the-art Dynamic Inverse Reinforcement Learning (DIRL) framework presented by Ashwood et al. (NeurIPS 2022). Related results are shown in Section 5.2 (Page 6) and Appendix D.2 (Page 16). Our approaches outperform DIRL in predicting mice behavior (indicated by the log-likelihood on held-out trajectories, Figure 3B, and Figure 7A) and are able to provide interpretable reward functions (Figure 3D, Figure 7B).
>
> *”The fitted model is somewhat simplistic. Latent states are assumed to be multinomial or Markov, but the most plausible biological assumption would be that transitions between drives also interact with reward/satiety/recent history."*
> *”Is it possible to incorporate some recent reward history into the transition structure? Since the inference algorithm is EM, will any EM-compatible latent variable model work, in principle?”*
>
> Thank you for this great question. It is indeed a pertinent consideration to explore the utilization of a controlled latent state transition system, potentially incorporating external covariables, rather than the autonomous system currently under consideration in this paper. A promising avenue for future research could involve adopting a generalized linear model with inputs such as reward, satiety, recent history, etc., in lieu of the current generalized Bernoulli process or Markov process controlling the latent state transition dynamics. This adjustment would allow for the incorporation of the identification of potential external factors influencing intention transition dynamics. Furthermore, there is merit in exploring the feasibility of exerting control over the intention transition. Your suggestion is invaluable, and we will thoroughly explore these possibilities in our ongoing work.
>
> *”Where are the bottlenecks for the method in terms of inferential complexity? Is the limiting factor the IAVI regression (i.e., the size of the tabular problem) or the EM complexity?”*
>
> Thanks for your question. The computational complexity of IAVI is akin to that of standard value iteration, with the difference being an additive term for solving the SLE – a term that is often of minor importance given the usual predominance of the number of states over the number of actions. The primary contribution of Kalweit et al. was indeed the reduction in computational complexity relative to other IRL approaches, which was the reason for our selection of IAVI. The limiting factor is, therefore, rather identified as the complexity of the EM in the outer loop, although the computational demands of solving the MDP in the inner loop are of course inherently significant. Hence, the limiting part cannot be universally assigned without context; it depends on specific factors such as the size of the state space, the convergence rate of the EM, the convergence criteria of value iteration, and the complexity of computations within each E-step.
>
> Does this address your concerns?

---

> > ### Comment · Reviewer_o7yT · 2023-11-22
> >
> > I appreciate the authors' responses, and the additional experiment is of added value. I do have a follow-up question:
> >
> > > Our approaches outperform DIRL in predicting mice behavior (indicated by the log-likelihood on held-out trajectories, Figure 3B, and Figure 7A) and are able to provide interpretable reward functions (Figure 3D, Figure 7B).
> >
> > I assume this is because LMV-IVAI has a more concentrated prior than DIRL? That is, if I understand the authors' responses to myself and other reviewers, the key difference here is that DIRL assumes $z$ changes _within trajectories_ while LM-IVAI assumes the $z$'s are _constant_ for demonstrations. In that case, LM-IVAI (as a latent variable model) would be a special case of DIRL in which no transitions of the latent state occur?
> >
> > So is the improved performance of LMV-IVAI due to the more concentrated prior on the latents or something else?

---

> > > ### Author Response · Authors · 2023-11-22
> > >
> > > Thanks for your further question. It appears that our previous responses to other reviewers may not have sufficiently clarified certain aspects. In the additional experiment where we compare our approaches with DIRL, the $z$’s also change ***within trajectories*** as that in DIRL. This can be observed in Figure 3E and Figure 7C, where we present the temporal latent state transition trajectories.
> > >
> > > An essential characteristic of our approach is that we allow ***both*** latent state transition within an episode (i.e. continuous-time transition similar to DIRL, as shown in the second and third experiment) ***and*** between episodes (i.e. stay constant within one episode, as shown in the first experiment). This distinction represents a significant advantage of L(M)V-IQL over DIRL, where the former allows the user’s to choose which transition dynamics to consider according to different tasks/environments/etc., while the latter, however, only permits continuous time-varying reward functions.
> > >
> > > Does this clarify address your concerns?

---

> > > > ### Comment · Reviewer_o7yT · 2023-11-22
> > > >
> > > > Yes, I think so.

---

### Official Review · Reviewer_bHzC · 2023-10-31

**Soundness:** 3 good
**Presentation:** 2 fair
**Contribution:** 3 good
**Rating:** 5
**Confidence:** 3

**Summary:**

This paper proposes an expectation-maximization approach for multi-goal IRL based on the inverse Q-learning IRL method. The approach involves clustering trajectories into multiple intentions and independently solving the IRL problem for each intention. The authors evaluate their algorithm using both simulated experiments and real-world mice data.

**Strengths:**

The problem of multi-goal IRL is highly relevant and its applications to cognitive science have been sparse in the past. I therefore particularly appreciate the application to the cognitive science domain to interpret real mice data.

The introduction of a multi-goal approach based on the inverse Q-learning algorithm seems novel and holds the promise of potentially outperforming previous multi-goal IRL methods.

The paper is well-structured and easy to follow, contributing to its readability and comprehensibility.

The algorithm's application to real mice behavioral data demonstrates its practical applicability in real-world scenarios.

**Weaknesses:**

The most significant weakness of the paper is the lack of discussion regarding related work on multi-goal IRL. Despite the existence of numerous prior works in this field (e.g., [1-6]), the paper does not reference or discuss any of them. The absence of a comparison with existing methods raises questions about the true novelty and contribution of the proposed approach. The paper should explicitly highlight what sets its method apart and how it compares to the existing literature. Especially works [4-6] previously approached the multi-goal IRL problem with an expectation-maximization approach, even though inverse Q-learning was not used as backbone algorithm.

[1] Dimitrakakis, C., & Rothkopf, C. A. (2012). Bayesian multitask inverse reinforcement learning. In Recent Advances in Reinforcement Learning: 9th European Workshop (EWRL), pp. 273-284, Springer

[2] Gleave, A., & Habryka, O. (2018). Multi-task maximum entropy inverse reinforcement learning. arXiv preprint arXiv:1805.08882.

[3] Babes, M., Marivate, V., Subramanian, K., & Littman, M. L. (2011). Apprenticeship learning about multiple intentions. In Proceedings of the 28th international conference on machine learning, pp. 897-904

[4] Choi, J., & Kim, K. E. (2012). Nonparametric Bayesian inverse reinforcement learning for multiple reward functions. In Advances in neural information processing systems, vol. 25., pp. 305-313

[5] Michini, B., & How, J. P. (2012). Bayesian nonparametric inverse reinforcement learning. In Machine Learning and Knowledge Discovery in Databases: European Conference, (ECML PKDD), pp. 148-163, Springer

[6] Bighashdel, A., Meletis, P., Jancura, P., & Dubbelman, G. (2021). Deep adaptive multi-intention inverse reinforcement learning. In Machine Learning and Knowledge Discovery in Databases (ECML PKDD), pp. 206-221, Springer

**Questions:**

How does your approach compare to previous multi-goal IRL methods, especially those mentioned in the references [1-6]? It is crucial to provide a detailed comparison to establish the uniqueness and advantages of your proposed method in light of the existing literature.


-----
I appreciate the additional discussion of past work provided by the authors. For me, the main problem is still that this work is conceptually very close to past multi-objective IRL approaches. Finally, they have added a discussion of related work, but they still claim their model with latent intention states and EM as their new contribution. It seems to me that their approach is basically the same as older work (after careful rereading, the closest is probably [7]), but the original IRL approach was swapped out for IAVI or IQL to support non-linear reward functions and have a model-free variant. I do not think there's anything wrong with that, and I think work showing that this combination works could still make a good paper and be useful to the community. However, I would have liked to discuss these close connections to other approaches with the authors to assess the actual novelty and ensure that the work is not overstated. I know that there is limited time in the rebuttal phase to make improvements, and they used that time well to add a schematic overview of previous algorithms. However, I do not like that they initially did not address past methods at all and provided this discussion on the last day of the rebuttal phase, which made a real discussion and improvement of the paper impossible. Therefore, I would suggest the authors make their paper clearer in terms of its true novelty and resubmit it so that a discussion with the reviewers can take place. I will still increase my rating to a 5.

[7] Nguyen, Q. P., Low, B. K. H., & Jaillet, P. (2015). Inverse reinforcement learning with locally consistent reward functions. In Advances in neural information processing systems, 28.

---

> ### Author Response · Authors · 2023-11-22
>
> We thank reviewer bHzC for his suggestions to improve our submission. In response to your concerns and questions, we have repositioned the section on related work, moving it from the Appendices to the primary body of the paper immediately following the introduction (Section 2, Page 2). To ensure a more comprehensive understanding and foster ease of comparison, we have incorporated a table summarizing key features of the relevant algorithms, including but not limited to the aforementioned references [1-6], as well as a detailed comparison of the advantages and uniqueness of our approaches. In brief, our algorithms are capable of estimating ***non-linear***, ***discrete time-varying*** reward functions within both ***model-based*** and ***model-free*** conditions.

---

### Official Review · Reviewer_M3j1 · 2023-11-01

**Soundness:** 3 good
**Presentation:** 2 fair
**Contribution:** 2 fair
**Rating:** 5
**Confidence:** 3

**Summary:**

This method, LMV-IQL, seeks to extend a class of IRL algorithms to the case of multiple intrinsic rewards, applied to behavioral modeling in neuroscience. They first identify each intention / reward and then solve for each. They demonstrate their method on both simulated and experimental datasets.

**Strengths:**

The text is well written, particularly when defining theorems and the algorithmic steps. It would be a definite strength to extend IRL approaches to the regime of multiple unknown (intrinsic) reward functions or internal motivation states.

**Weaknesses:**

Some of the figures need additional details/components. E.g., Figure 1, 2 need color scalebars, Figure 3 would benefit from some explanation of the legend (where are the red and blue squares?), the colors on the state labels in Figure 4C are unnecessary and uncorrelated with the colors in the legend, ...

Definitions of the comparison methods were weak. For example, 'IAVI was further extended to the sampling-based model-free Inverse Q-learning (IQL) algorithm' with no citation or explanation of how the authors of this paper implemented those algorithms, is insufficient.

Similarly, the primary metric, EVD, is cited but not defined.

The authors only show an improvement over IAVI and IQL, and do not compare these other methods (including LV-IQL, which performed the same on the simulated dataset) in the experimental dataset.

**Questions:**

The authors motivate their method as extending beyond a single reward function, but then apply it only to the case of 2-3 rewards/intentions/states. Can this be extended easily to more than a small number?

---

> ### Author Response · Authors · 2023-11-22
>
> We thank reviewer M3j1 for his suggestions to improve our submission. We address your concerns in detail below and have updated our paper accordingly.
>
> *“Some of the figures need additional details/components. E.g., Figure 1, 2 need color scalebars, Figure 3 would benefit from some explanation of the legend (where are the red and blue squares?), the colors on the state labels in Figure 4C are unnecessary and uncorrelated with the colors in the legend, …”*
>
> Thanks, we have updated the figures.
>
> *“Definitions of the comparison methods were weak. For example, 'IAVI was further extended to the sampling-based model-free Inverse Q-learning (IQL) algorithm' with no citation or explanation of how the authors of this paper implemented those algorithms, is insufficient.”*
>
> Thanks for your suggestion. IQL was initially introduced alongside IAVI as a model-free extension in Kalweit et al. (2020), and we have faithfully implemented their approach in our current work without any modifications. In response to your feedback, we have diligently revised the relevant sections of our paper, providing more explicit citations for clarity.
>
> *“Similarly, the primary metric, EVD, is cited but not defined.”*
>
> Thanks, we have updated the EVD definition in our paper for better clarity.
>
> *“The authors only show an improvement over IAVI and IQL, and do not compare these other methods (including LV-IQL, which performed the same on the simulated dataset) in the experimental dataset.”*
>
> Thanks for your great suggestions. We included an additional experiment on real mice navigation trajectories within a labyrinth from Rosenberg et al. (2021) as a benchmark to compare our algorithm with the state of the art — Dynamic Inverse Reinforcement Learning (DIRL) from Ashwood et al. (NeurIPS 2022). Related results are shown in Section 5.2 (Page 6) and Appendix D.2 (Page 16). Our approaches outperform DIRL in predicting mice behavior (indicated by the log-likelihood on held-out trajectories, Figure 3B, and Figure 7A) and are able to provide interpretable reward functions (Figure 3D, Figure 7B).
>
> *“The authors motivate their method as extending beyond a single reward function, but then apply it only to the case of 2-3 rewards/intentions/states. Can this be extended easily to more than a small number?”*
>
> Thank you for your insightful question. While our manuscript predominantly delves into a detailed exploration of the learned reward function and intentions from LMV-IQL with 2-3 latent states, we also conducted model fitting with a 4-latent-state LMV-IAVI on mice navigation trajectories and a 5-latent-state LMV-IQL on mice reversal-learning behavior. The primary practical constraint of EM based algorithms arises from computational time, particularly as the number of intentions increases—an inherent limitation shared with EM-MLIRL (Babes et al., 2011) and MI-Σ-GIRL (Likmeta et al., 2021). Notably, IAVI possesses the unique capability to ***analytically*** derive a matching reward function for observed behavior in ***closed-form***, thus omitting the computationally intensive inner loop at each iteration of EM. This analytical feature distinguishes our approaches, enabling their application on more expansive latent state spaces compared to existing algorithms.
>
> Does this address your concerns?

---

> > ### Comment · Reviewer_M3j1 · 2023-11-23
> >
> > I have read the revised manuscript and the other reviews and responses. My concerns about sufficient comparisons with SOTA are somewhat mitigated by the new experiment shown in section 5.2. I will raise my score from a 3 to a 5.

---

### Author Response · Authors · 2023-11-22
**Overview of Revision**

We thank all reviewers for their constructive feedback and for helping us make our paper a stronger submission! We have uploaded a revised version of the paper with the major changes highlighted in blue. The main issues of the reviewers which we have addressed in the revised manuscript are:

* **Introduction:** In order to enhance the lucidity of our algorithm's underlying motivation, we have incorporated a succinct introduction to the current leading framework in animal behavior characterization—Dynamic Inverse Reinforcement Learning (DIRL) as presented by Ashwood et al. (NeurIPS 2022).
* **Related work:** Recognizing the importance of accessibility, we have relocated the section on related work from the Appendices to the main body of the paper, after the introduction. In order to facilitate a more comprehensive understanding and facilitate ease of comparison, we have introduced a table summarizing key features of the related algorithms. Moreover, we have expanded our comparative analysis by incorporating additional insights from prominent references, including but not limited to:
    * Dimitrakakis, C., & Rothkopf, C. A. (2012). Bayesian multitask inverse reinforcement learning. In Recent Advances in Reinforcement Learning: 9th European Workshop (EWRL), pp. 273-284, Springer
    * Gleave, A., & Habryka, O. (2018). Multi-task maximum entropy inverse reinforcement learning. arXiv preprint arXiv:1805.08882.
    * Michini, B., & How, J. P. (2012). Bayesian nonparametric inverse reinforcement learning. In Machine Learning and Knowledge Discovery in Databases: European Conference, (ECML PKDD), pp. 148-163, Springer
    * Bighashdel, A., Meletis, P., Jancura, P., & Dubbelman, G. (2021). Deep adaptive multi-intention inverse reinforcement learning. In Machine Learning and Knowledge Discovery in Databases (ECML PKDD), pp. 206-221, Springer.
* **Algorithms:** In consideration of spatial constraints, we have elected to relocate the pseudocode corresponding to Algorithm 1 from the main body of the paper to Appendix B.1 (Page13).
* **Experiments**
    * **Gridworld experiment**
        * To enhance the clarity of presentation, we have omitted the figure depicting t-SNE visualization of trajectories (Figure 3 from the prior revision). Instead, pertinent results are now depicted in Figure 2 (Top), where trajectories are directly plotted within the Gridworld environment, with the corresponding state-values displayed in the background.
        * An additional experiment within the same Gridworld environment configuration was undertaken. We removed the punishment associated with intention-irrelevant rewards, replacing the previously assigned −1 reward on intention-irrelevant states with a neutral value of 0. Comprehensive results pertaining to this modification are elucidated in Appendix C.2 (Page 15).
    * **New additional experiment as benchmark for algorithm comparison: In order to bolster the robustness of our findings, we extended our experimental scope to include an investigation of real mice navigation trajectories within a labyrinth from Rosenberg et al. (2021) with LMV-IAVI. This expansion served as a benchmark, allowing for a comparative evaluation against the state-of-the-art Dynamic Inverse Reinforcement Learning (DIRL) framework presented by Ashwood et al. (NeurIPS 2022). Detailed results stemming from this comparative analysis are explicated in Section 5.2 (Page 6), and Appendix D.2 (Page 16).**
    * **Experiment on mice reversal-learning behavior**
        * In order to enhance the concise presentation of critical content and eliminate redundant figures, we have opted to exclude Figure 5D from the previous revision in the current iteration. Furthermore, we have repositioned the remaining figures originally found in Figure 4 and Figure 5 of the prior revision to now occupy Figure 4 and Figure 5 in the current revision.
        * We have refined the color palette and marker representation in Figure 4C from the prior iteration to enhance overall visual clarity. This enhancement is distinctly depicted in Figure 4D of the current revision.
        * We additionally computed the Bayesian Information Criterion (BIC) for LMV-IQL across a varying number of latent states. This analysis was conducted with the specific purpose of providing argumentations for our model selection in subsequent analysis.
* **Conclusion:** We have endeavored to streamline the conclusion section in comparison to the prior revision, aiming for increased clarity and conciseness. Additionally, we have introduced supplementary remarks on prospective avenues for future research in the last paragraph.

---

### Meta-Review · Area_Chair_xDSx · 2023-12-07

**Metareview:**

This paper considers inverse reinforcement learning for animal behavior data with multiple possible latent intentions. The approach uses expectation-maximization to cluster demonstrated behaviors and then learns separate reward functions for each cluster. The reviewers appreciated the motivations for considering multiple intentions, which were well-articulated. The effectiveness of this approach is illustrated on both synthetic data and mice behavior data. In the original submission, closely related work was not adequately discussed. Though the authors strengthened the paper by better situating the paper's technical contributions with respect to previous methods in their revision, there are remaining concerns about whether the novelty of the approach is sufficient for publication in ICLR. The novelty of the proposed approach needs to be better delineated from the previous works provided by the reviewers. This requires more substantial revision and prevents a recommendation for acceptance in its current form.

**Justification For Why Not Higher Score:**

Despite revisions, there are still concerns about the claims of novelty with respect to existing work.

**Justification For Why Not Lower Score:**

N/A

---

### Decision · Program_Chairs · 2024-01-16

Reject